# The Effect of (Poly)phenol-Rich Interventions on Cognitive Functions and Neuroprotective Measures in Healthy Aging Adults: A Systematic Review and Meta-Analysis

**DOI:** 10.3390/jcm9030835

**Published:** 2020-03-19

**Authors:** Achraf Ammar, Khaled Trabelsi, Patrick Müller, Bassem Bouaziz, Omar Boukhris, Jordan M Glenn, Nick Bott, Tarak Driss, Hamdi Chtourou, Notger Müller, Anita Hökelmann

**Affiliations:** 1Institute of Sport Sciences, Otto-von-Guericke University, 39104 Magdeburg, Germany; anita.hoekelmann@ovgu.de; 2UR15JS01: Education, Motricity, Sport and Health (EM2S), High Institute of Sport and Physical Education, University of Sfax, Sfax 3000, Tunisia; trabelsikhaled@gmail.com (K.T.); omarboukhris24@yahoo.com (O.B.); 3High Institute of Sport and Physical Education, University of Sfax, Sfax 3000, Tunisia; 4German Center for Neurodegenerative Diseases (DZNE), 39120 Magdeburg, Germany; patrick.mueller@dzne.de (P.M.); notger.mueller@dzne.de (N.M.); 5Department of Neurology, Medical Faculty, Otto von Guericke University, 39104 Magdeburg, Germany; 6Higher Institute of Computer Science and Multimedia of Sfax, University of Sfax, Sfax 3000, Tunisia; bassem.bouaziz@isims.usf.tn; 7Exercise Science Research Center, Department of Health, Human Performance and Recreation, University of Arkansas, Fayetteville, AR 72701, USA; jordan@neurotrack.com; 8Neurotrack Technologies, 399 Bradford St, Redwood City, CA 94063, USA; nick@neurotrack.com; 9Clinical Excellence Research Center, Department of Medicine, Stanford University School of Medicine, Stanford, CA 94305, USA; 10Interdisciplinary Laboratory in Neurosciences, Physiology and Psychology: Physical Activity, Health and Learning (LINP2-2APS), UFR STAPS, UPL, Paris Nanterre University, 92000 Nanterre, France; tarak.driss@parisnanterre.fr; 11Physical Activity, Sport and Health, UR18JS01, National Observatory of Sport, Tunis 1003, Tunisia; h_chtourou@yahoo.fr

**Keywords:** cognition, neuroplasticity, neuroinflammation, brain, polyphenols, meta-analysis

## Abstract

Context: As the food industry is continually involved in the development of new attractive alternative therapeutic agents, the evaluation of the beneficial impact of (poly)phenols on cognitive and brain function during aging has gained increasing interest. Objective: This systematic review and meta-analysis aimed to evaluate the acute and chronic effects of (poly)phenol-rich diet supplementation on cognitive function and brain health in aging adults. Data Sources: PubMed and Web of Science databases were searched for relevant randomized placebo-controlled trials (RCTs) published from inception to July 2019. Study Selection: Two researchers independently screened 4303 records, using the PICOS criteria: Participants were aging adults; Intervention was based on acute and/or chronic (poly)phenols-rich supplementation; Comparator was any comparator; Outcomes included cognitive function and neuroprotective measures; and Study design was RCTs. A third researcher was consulted when discrepancies arose. Fifteen high-quality (mean PEDro score = 8.8 ± 0.56) RCTs (total participants: 918 healthy older adults) were included in the final sample. Data Extraction: Information on study design, employed treatment, characteristics of participants, outcomes, and the correspondent assessing methods were extracted. Preferred Reporting Items for Systematic Reviews and Meta-Analyses (PRISMA) guidelines were followed. Data Analysis and Results: A random-effects meta-analysis was used to pool estimates across studies. Effect size (ES) and its 95% confidence interval (CI) was calculated. Pooled results yielded a trivial ES (−0.2 to 0.03) for brain-derived neurotrophic factor and neuroinflammatory parameters and small (0.36) to moderate (0.82) ES for executive functions. Conclusion: This meta-analysis failed to provide evidence regarding the neuroprotective and anti-inflammatory effect of (poly)phenols supplementation in aging adults. However, findings from individual studies, included in this systematic review, suggest polyphenol-rich supplementation may improve some cognitive and brain functions in older adults. The beneficial effect of polyphenols seems to depend on ingested dose and bioavailability. Results suggest at least an intermediate dose (≥500 mg), and intermediate (≈9%) to high (43%) bioavailability rates are needed to cross the brain blood barrier and to exert a significant effect on cognitive health.

## 1. Introduction.

For the first time in history, most people can expect to live into their sixties and beyond [1]. Additionally, by 2050, the proportion of the world’s population over 60 years will be expected to nearly double from currently 12% to 22% [1]. While this increase in average life expectancy is one of society’s great achievements, the resulting demographic shift has brought with it a growing number of chronic diseases such as diabetes, cancer, cardiovascular diseases, and neurodegenerative diseases [1]. Brain aging, the key risk factor for neurodegeneration [2], is a highly complex biological process that is inexorably associated with a more or less severe age-related cognitive decline (ARCD) [3]. Among ARCD, declines in memory, executive functioning, attentional capacities, and processing speed are the most serious brain-health concerns for the elderly [4,5]. The etiology of these age-associated cognitive losses is complex and multifactorial [6,7,8]. One set of ARCD’s factors seems related to oxidative stress, leading to neuroinflammatory process in neuronal cells [7]. Indeed, given that cognitive impairment and oxidative stress are common occurrences in old age, multiple studies have identified oxidative stress as a causative link between normal brain aging and various neuropathological conditions [7,9]. The redox imbalance in favor of pro-oxidant in older adults is mainly due (i) to the increased generation of reactive oxygen species because of the higher accumulation of reactive iron with aging [10,11] and (ii) to the decreased activity of many antioxidant enzymes (i.e., superoxide dismutase, catalase, glutathione, and glutathione peroxidase) in specific cognitive regions of the brain such as the hippocampus and the cerebral cortex [12,13]. This imbalance reduces the brain’s capability to neutralize/counter the generated free radicals (FR) and, consequently, increases vulnerability to FR attack, potentially leading to tissue damage and neuroinflammatory processes [14,15]. This enhanced neuroinflammatory process may cause dysfunction and/or death of neurons, eventually resulting in cognitive decline [16,17]. Additionally, this may play an important role in the apparition of neurodegenerative disease via toxin generation pathways [9,18].

Given the age-dependent inability to counterbalance the altered redox status by endogenous defenses (e.g., enzymatic antioxidant), it is imperative to develop drugs and/or lifestyle interventions (e.g., physical exercise) that can catalyze antioxidant and neuroprotective actions in order to slow, prevent, or even reverse age-related cognitive disorders. One such possibility is the use of potent antioxidant nutritional substances such as (poly)phenols [19,20]. These (poly)phenol compounds (i.e., flavones, flavonols, isoflavones, flavanones/flavanonols, flavanols, and anthocyanidins) are present in high amounts in fruits, vegetables, and natural products such as parsley, celery, onions, leeks, broccoli, soy, oregano herbs, green tea, red wine, citrus fruit, berry fruits, and dark chocolate [20,21]. Although research has historically focused on antioxidant properties [22,23,24,25], recent data from randomized placebo-controlled trials suggest (poly)phenols may also modulate neurological disorders, cerebral hypoperfusion, and neuroinflammation while simultaneously enhancing memory, learning, and cognitive performances in old-aged adults [19,26,27,28,29,30,31,32,33,34]. Mastroiacovo et al. [26] demonstrated a daily consumption of 250 mL Cocoa drinks, with a high flavanol content (993 mg), for 8 weeks has the potential to improve psychomotor performance, verbal fluency performance, as well as the overall cognitive performance (i.e., z-score). Whyte et al. [27] indicated that 3 months of daily supplementation with 100 mg purified wild blueberry extract, with a total (poly)phenol of 50 mg, improved both verbal and short-term spatial episodic memory functions. Furthermore, Wightman et al. [28] showed that acute supplementation with Greek mountain tea (950 mg) improves accuracy of working and episodic memory, speed of attention, processing of visual information, and cerebral hemodynamics in the prefrontal cortex. Improvements in resting regional cerebral blood flow have also been reported following an acute ingestion of high-flavanol (494 mg) cocoa drink [30] and after daily ingestion of 387 mg anthocyanin-rich blueberry concentrate for 12 weeks [31]. More interestingly, an ongoing (1997–present) community-based, prospective cohort of 921 older persons showed that higher dietary intakes of flavanols may be associated with reduced risk of developing Alzheimer dementia [34]. However, other studies have reported nonsignificant effects or even unwanted effects of (poly)phenol-rich supplementation on certain cognitive functions, specifically executive functioning, working memory and verbal memory [35,36], or cerebral blood flow response [37].

Randomized placebo-controlled trials (RCTs) are the gold standard to confirm the effects of nutritional interventions (e.g., rich-(poly)phenol supplementation) on cognitive decline, maintenance, or improvement [38]. However, the aforementioned trials have been limited by sample size, supplement dose, and research design and seem underpowered to achieve a comprehensive and reliable conclusion. Meta-analysis provides an opportunity to overcome this limitation by increasing the sample size. The present study aimed to resolve this uncertainty by systematically reviewing the literature and by conducting a meta-analysis of all trials investigating the acute and chronic effects of (poly)phenol-rich supplementation on cognitive functions and brain health in old-aged adults.

## 2. Method

The systematic review and meta-analysis were conducted and reported in accordance with the guidelines of the preferred reporting items for systematic reviews and meta-analysis (PRISMA), an evidence-based protocol describing criteria for reporting in systematic reviews and meta-analyses [39].

### 2.1. Data Sources and Search Strategy

To inform our review, a comprehensive systematic search of studies was performed electronically in PubMed/Medline and Web of Science databases considering all manuscripts published in the English language from inception to July 2019. Search terms (including mapping to appropriate Medical Subject Headings (MeSH) terms where appropriate) described major (poly)phenol classes (e.g., polyphenol, flavonoids, polyphenolic compounds, isoflavone, flavanol, resveratrol, etc.) in combination with keywords relating to cognitive and brain functions (e.g., cognitive performance, cognitive function, neuroimaging, brain volume, brain structure, Brain-derived neurotrophic factor (BDNF), regional perfusion, etc.). The search was conducted with additional filters excluding nonhuman studies and studies in diseases (see Appendix A online for the full search strategy) and was applied to titles and abstracts. To identify additional studies not included in these search terms, the search was supplemented by manually cross-matching reference lists, key author searches, and citation searching of relevant research and review articles. The search strategies were combined, and duplicates were removed by Endnote and manually checked by two of the authors. Once all relevant articles were located, the researchers individually considered each article for its appropriateness for inclusion based on the predetermined inclusion criteria described below. Where there was uncertainty with regard to inclusion, discussion with a third researcher determined the final inclusion or exclusion of the article.

### 2.2. Inclusion and Exclusion Criteria

To be included in the systematic review, each individual study was required to meet the following inclusion criteria: (i) primary research published in peer-reviewed journals in English, (ii) research conducted with healthy aging human adults aged 55 years old and over, (iii) original studies investigating an acute or chronic(poly)phenol-rich supplementation intervention on cognition and brain-related functions, (iv) no severe methodological deficiencies (e.g., allocation not randomized, absence of control comparison (e.g., Placebo (PLA) or very low poly(phenols) dose and/or content, etc.), participant not blinded, and inappropriate statistical analysis procedures), and (v) published before August 2019. Exclusion criteria were (i) studies written in languages other than English, (ii) data from congress or workshop publications, (iii) animal studies, (iv) studies in which no supplementation was given, (v) studies which administered multiple supplements in addition to (poly)phenol, (vi) studies conducted with participants from different age ranges (e.g., tested population include both young and older adults), and (vii) studies conducted in aging populations with current cognitive impairment/dysfunction or other diseases. Case studies, encyclopedias, book chapters, and reviews were excluded, although the bibliographies of the latter were consulted to refine article searches.

### 2.3. Study Selection

Following the removal of duplicate studies from the different search engines, inclusion or exclusion of the remaining articles was performed by applying the aforementioned criteria on the title and abstract to determine eligibility in a preliminary independent screening. Selected papers were then read in full to finalize eligibility in accordance with the PICOS (population, intervention, comparison, outcome, and study design) criteria shown in Table 1. A summary of the study selection process is outlined in Figure 1. The university’s library, hand searches, electronic databases, and contact with the authors were used to obtain full copies of the published manuscripts.

### 2.4. Data Extraction

Data were extracted using a standardized form. The following data were extracted from each study: primary author’s first name, year of publication, study design, treatment characteristics, dosage of supplements, characteristics of the treatment and placebo groups, and intervention duration. The primary outcomes extracted in this review were the acute and/or chronic effects of (poly)phenol-rich supplementation on a range of cognitive functions such as psychomotor function (e.g., reaction time), visual processing, attention, executive function, verbal and spatial memory, working memory, learning ability, and other specific cognitive functions assessed using validated neuropsychological measures. These outcomes are presented in Table 2. All data concerning the acute and/or chronic effect of (poly)phenol-rich supplementation on neuroimaging, cerebral blood flow, cerebral hemodynamics, neuroplasticity, neuroinflammation, and other brain-related parameters (i.e., near-infrared spectroscopy (NIRS), transcranial doppler ultrasound (TCD), functional magnetic resonance imaging (fMRI), or blood analysis) were extracted from the research papers and are shown in Table 3. For all extracted outcomes, information regarding the assessment methods (e.g., cognitive test battery and neuroimaging techniques) are provided in specific columns within both tables.

### 2.5. Quality Assessment

To assess the methodological quality of the selected studies, the Physiotherapy Evidence Database (PEDro) scale was used [22,46]. The PEDro scale is based on the Delphi list developed by Verhagen and colleagues at the Department of Epidemiology, University of Maastricht [47]. The PEDro scale is a reliable and objective tool that helps identify which of the randomized controlled trials from the same areas of physiotherapy practice are likely to be externally (criteria 1) and internally (criteria 2 to 9) valid and could have sufficient statistical information to make their results interpretable (criteria 10 and 11) [22,47]. Each paper was independently assessed twice by two independent reviewers using the 11-item checklist to yield a maximum score of 10 (the sum of awarded points for criteria 2 to 11). Points are only awarded when a criterion is clearly satisfied and when criterion one, which relates to external validity, is not used to calculate the PEDro score. Each manuscript was assessed by two of the four authors, and discrepant results were resolved through a consensus meeting. From previous studies [22,48], a score of 9–10 on the PEDro scale was considered “high quality”, scores of 5–8 were considered “moderate quality”, and studies that scored below 5 were considered “low quality”.

### 2.6. Statistical Analysis

Meta-analysis was conducted using the commercial software “Comprehensive Meta-Analysis” (CMA for Windows, version 3, Biostat, Englewood, NJ 2013, USA). Given the high variability in cognitive tasks between the included studies, only executive functions during Trail Making Test A (TMT-A) and B (TMT-B) showed to be sufficiently comparable and were included in the meta-analysis. Similarly, given the high variety of the assessed brain-related parameters as well as the diversity in measurement techniques (i.e., MRI, fMRI, transcranial Doppler sonography (TCD), Near-infrared spectroscopy (NIRS), and blood analysis), the meta-analysis was only conducted for Brain-derived neurotrophic factor (BDNF) as a biomarker of neuroplasticity and for HsCRP, IL-6, and TNF-α as biomarkers of neuroinflammation. To calculate the effect size, performance in TMT-A and TMT-B were collected in seconds (s), BDNF and Hs-CRP blood concentrations were collated in ng/mL and mg/L respectively, and blood concentrations of IL-6 and TNF-α were collected on pg/mL. In studies where net changes were not directly reported in the intervention and control groups, the effect size was computed by subtracting the values at the endpoint of the intervention from those at baseline. The standard deviations of mean differences were calculated by using SD = square root ((SD pretreatment)2 + (SD posttreatment)2 − (2R × SD pretreatment × SD posttreatment)), with the correlation coefficient (R) assumed to be 0.5 [49,50]. Effect size (ES) and its 95% confidence interval (CI) were calculated utilizing Cohen’s method, reflecting the standardized difference in means (SDM) between measured parameters (i.e., TMT-A, TMT-B, BDNF, Hs-CRP, IL-6, and TNF-α), both in response to (poly)phenol-rich supplementation and to placebo. ES was interpreted as trivial (ES < 0.2), small (ES between 0.2 and 0.6), moderate (ES between 0.6 and 1.2), large (ES between 1.2 and 2.0), very large (ES > 2.0), and extremely large (ES > 4.0) [51]. A positive ES value in BDNF indicated that (poly)phenol-rich supplementation increased outcomes, while a negative ES in the remaining parameters indicated (poly)phenol-rich supplementation decreased outcomes. Q and *I*^2^ statistics were utilized to assess statistical heterogeneity [52,53]. Substantial heterogeneity was considered for *I*^2^ value >50% and indicated that a random-effect model was preferred to a fixed-effect model [53]. Funnel plots’ potential asymmetries, the Begg and Mazumdar’s rank correlation test (Kendall’s S statistic P-Q) [54], the Egger’s linear regression test [55], and the Duval and Tweedie’s trim-and-fill test [56] were used to examine publication bias. The stability of the pooled ES of each study was assessed via sensitivity analyses by removing individual studies from the analysis and by computing the impact of the excluded study. A cumulative meta-analysis was realized to further ensure the stability and reliability of the results.

## 3. Results

Fifteen studies [26,27,28,29,30,31,35,36,37,40,41,42,43,44,45] met the inclusion criteria and were included in the current systematic review. The studies examined the effects of (poly)phenol-rich supplementation intake on cognitive functions and/or brain-related parameters (e.g., neuroimaging, neuroplasticity, CBF, etc). All studies used a statistical significance threshold of *p* < 0.05.

### 3.1. Study Selection and Characteristics

#### 3.1.1. Study Selection

The predefined search strategies yielded a preliminary pool of 4303 possible papers. Removal of duplicates resulted in a selection of 2615 published papers. Removal of nonclinical trial resulted in a selection of 230 published papers. A first screening of titles and abstracts for eligibility against inclusion and exclusion criteria led to a provisional list of 38 published studies. After a careful review of the 38 full texts, 23 articles were excluded (16 studies investigated only young and/or middle-aged adults and did not investigate the old-age population and 7 studies investigated a heterogeneous population including both young and older adults in the same group). Therefore, 15 studies met the established inclusion criteria for determining the effects of (poly)phenol-rich supplementation on cognitive functions and a variety of neurological related outcome measurements among aging adults. A summary of this process can be seen in Figure 1.

#### 3.1.2. Study Characteristics

The characteristics of each study as well as the cognitive and the neurological changes following rich-(poly)phenol supplementation compared to placebo supplementation are summarized in Table 2 and Table 3, respectively. Eight papers [26,27,35,36,40,41,43,44] examined only the effect of rich-(poly)phenol supplementation on different cognitive functions such as visual attention, working memory, reaction time, executive functioning, and learning abilities. Five studies [28,29,31,37,42] examined the effect of rich-(poly)phenol supplementation on cognitive function as well as a variety of brain-related parameters such as brain perfusion, cerebral blood flow, cerebral hemodynamics, hippocampal functional connectivity, neuroplasticity, neuroinflammation, and other brain-related parameters. Two studies [30,45] only examined the change in cerebral blood flow parameters following rich-(poly)phenol supplementation without cognitive function measurements. Concerning the acute (up to 2 h) and chronic effects of (poly)phenol-rich supplementation on the abovementioned functions, twelve studies investigated the chronic effect [26,27,29,31,35,36,40,41,42,43,44,45], two studies investigated only the acute effect [30,37], and only one study [28] investigated both acute and chronic effect of (poly)phenol-rich supplementation on cognitive and brain functions.

### 3.2. Subject Characteristics

The studies in this systematic review included a total of 918 participants. In studies employing a within-subject counterbalanced design, the number of participants in each trial ranged from 12 [37] to 30 [36] while studies employing a parallel groups design ranged from 21 [45] to 140 [28]. These studies targeted healthy aging adults with mean age ranging from 57 [37] to 74 [35] years.

### 3.3. Study Design and Supplement Administration

As presented in Table 2 and Table 3, all reviewed studies employed a randomized design. Eight studies employed two parallel experimental arms, with seven of them using placebo and rich-(poly)phenol supplementation as treatment arms [29,31,35,41,42,43,44], while the remaining study [45] used poor- and rich-(poly)phenol supplementation as treatment arms. Two studies employed three parallel experimental arms with one study using placebo as the control arm and low and high (poly)phenol doses as treatment arms [40] and the second using only (poly)phenol supplementation at different doses (i.e., low, moderate, and high) [26]. Two studies used four parallel experimental arms with one arm being the placebo and three arms for the different doses of (poly)phenol supplementation [27,28]. Three studies used one experimental arm (i.e., within-subject counterbalanced design), with a 1-week [30,37] or a 4-week washout period [36]. The majority of these studies (twelve out of fifteen) implemented a double-blind, placebo controlled experimental design. The three remaining studies focused on the effect of different doses of (poly)phenol supplementation without using a placebo-control: two studies implemented a double-blind design [26,45] and one study implemented a single-blind design [37]. The fifteen trials included in this review employed different varieties of dietary (poly)phenol supplementation with an intervention period that ranged from acute (up to 2 h) to multiple weeks/months (i.e., 2 weeks to 6 months). Three studies [29,40,42] opted for resveratrol extract treatment with a dose ranging from 200–300 mg/day (low–moderate dose) to 1000 mg/day (high dose). Three studies [35,36,43] opted for isoflavones extract treatment with a dose ranging from 55 mg/day to 110 mg/day (high dose). Three studies opted for blueberry-based extract treatment that allowed for a total daily (poly)phenol dose ranging from 35 mg to 70 mg in the study of Whyte et al. [27], a daily flavonoids dose of 258 mg in the study of Bensalem et al. [41], or a daily anthocyanidins dose of 387 as reported in the study of Bowtell et al. [31]. Four studies opted for rich-flavanol cocoa treatment that allowed for a total daily (poly)phenol dose ranging from 85 mg to 395 mg in the study of Marsh et al. [37] or a daily flavanol dose ranging from 29 mg (low dose) to 993 mg (high dose) as reported in the studies of Lamport et al. [30] and Sorond et al. [45]. The two remaining studies opted for flavonoid-rich Ginkgo biloba extract treatment with a daily dose of 180 mg [44] or a polyphenol-rich Greek mountain tea with a daily dose of 475 or 950 mg [28] without delineating the exact the polyphenol content in each dose. Additionally, different cognitive test batteries such as the Cambridge Neuropsychological Test Automated Battery (CANTAB) Paired Associate Learning test [41], the cognitive battery of tests (CogStateLtd.) [31,37], the cognitive function test battery (www.cognitivetesting.co.uk) [28], or a combination of validated cognitive tests [26,27,29,35,36,40,42,43,44] have been employed to assess the effect of (poly)phenol-rich supplementation on a variety of cognitive functions in aging populations, as presented in Table 2. Similarly, different neuroimaging techniques such as MRI [29,30,31,42], transcranial doppler (TCD) ultrasonography [37,45], and near-infrared spectroscopy (NIRS) [28] have been employed to assess the effect of polyphenol-rich supplementation on a variety of neurological functions in aging populations, as presented in Table 3.

### 3.4. Methodological Quality of Studies

All reviewed studies received a high score of seven and above with a mean PEDro score of 8.8 ± 0.56. Of the 15 studies included, 13 investigations received a very high score of 9 (i.e., authors employed a double-blind but not triple-blind trial), 1 investigation [40] scored 8 out of 10 as the authors failed to conceal allocation and to employ a triple-blind trial (i.e., double blind design was employed), and 1 investigation [37] scored 7 out of 10 as the authors failed to conceal allocation and to blind therapists and investigators. Overall, the study quality was deemed to be good to excellent (Table 4).

### 3.5. Effect of (Poly)Phenol-Rich Supplementation on Cognitive Functions of Older Adults

Of the 15 studies included in this meta-analysis, 13 assessed the acute and/or chronic effect of (poly)phenol-rich supplementation on cognitive functions of aging adults (Table 2). Two studies showed no significant effect of acute [37] or chronic [42] (i.e., 26 weeks) administration of (poly)phenol-rich supplementation (cocoa-rich chocolate with 85–395 mg polyphenol [37] or 200 mg resveratrol/day [42]) on cognitive function (e.g., memory, visual attention, learning ability, pattern recognition, etc.). Six studies showed that chronic ingestion (6 to 26 weeks) of (poly)phenol-rich supplementation resulted in a significant improvement of only one cognitive function out of the overall tested cognitive functions.

(i) A dose of 1000 mg resveratrol/day consumed over a 90-day period showed improved information processing time on the TMT-A with no effect on the other assessed cognitive functions (visual attention, working memory, verbal fluency, and semantic memory) [40].

(ii) A dose of 258 mg flavonoids/day over a 24-week period improved episodic recall memory on the verbal free recall memory (VRMFR), with no effect on visuospatial learning, recognition, and working memories assessed using CANTAB battery test [41].

(iii) A daily dose of 387 mg anthocyanins over a 12-week period significantly improved working memory performance during a 2-back cognitive task with nonsignificant effects on psychomotor function, visual processing, executive function, attention, verbal learning, and delayed record assessed using the “CogState Ltd.” cognitive test battery [31].

(iv) A daily dose of 110 mg isoflavones demonstrated improved verbal memory performance during a category fluency test but not immediate and delayed verbal recall memory and visuomotor tracking and attention functions during TMT-A and TMT-B [43].

(v) A daily dose of 180 mg flavonoid-rich Ginkgo biloba extract EGb 761 for 6 weeks significantly improve speed of processing abilities during only the Stroop-color-naming task with no significant effect during TMT-A, TMT-B, and Wechsler memory scale [44].

(vi) A dose of 200 mg resveratrol/day showed to significantly improve retention ability with nonsignificant effects on delayed recall, recognition, and learning abilities [29].

Of the reviewed 13 studies in Table 2, only 3 studies showed significantly enhanced performance following the chronic (1–6 months) and/or acute consumption of (poly)phenol-rich supplementation on at least two cognitive functions. Mastroiacovo et al. [26] showed that a daily consumption of 250 mL cocoa drink with high (993 mg) or moderate (520 mg) total flavanols content for 8 weeks may improve executive functions during TMT-A and TMT-B tests as well as the overall cognitive performance “z score.” Whyte et al. [27] showed that 3 months of daily supplementation with 100 mg purified wild blueberry extract with a total polyphenol of 50 mg improved both verbal and short-term spatial episodic memory functions but with no effect on working memory and executive functions, increasing the supplementation period to up to 6 months blinded the beneficial effect of polyphenols on all tested cognitive functions. Wightman et al. [28] showed that both acute and chronic (4 week) supplementation of Greek mountain tea improved accuracy of both working and episodic memory using the intermediate (475 mg) and high tea doses (950 mg) as well as speed of attention (i.e., reaction time) and processing of visual information. The remaining two studies (out of the 13 studies reviewed in Table 2) showed either an enhancement (e.g., verbal and visuospatial memory), a stability (e.g., verbal learning, paragraph recall, and language) or a decrease (e.g., digit recall and executive functions during Stroop color word and TMT-B tests) of cognitive functions following 6 months of daily isoflavones-rich supplementation (100 mg/day [35] or 55 mg/day [36])

### 3.6. Effect of (Poly)Phenol-Rich Supplementation on Brain Parameters of Older Adults

A total of seven studies assessed the acute and/or chronic effect of (poly)phenol-rich supplementation on brain parameters of aging adults (Table 3). Two studies focused on resting brain perfusion, showing beneficial effects of an acute ingestion of high-flavanol (494 mg) cocoa drink on resting regional CBF in the anterior cingulate cortex and central opercular cortex [30] as well as a beneficial effect of a 12-week daily ingestion of 387 mg anthocyanin-rich blueberry concentrate on regional grey matter perfusion in the parietal and occipital lobes [31]. Two studies focused on hippocampal volumetry and connectivity, showing that a daily intake of 2000 mg/day for 26 weeks increased hippocampal functional connectivity [29] with no effect on total gray matter, hippocampal volumes, or mean weighted hippocampal microstructure [29,42]. Two studies focused on cerebral blood flow responses to acute and/or chronic (poly)phenols-rich supplementation in older adults showing that 2 weeks of flavanol-rich (300 mg/day) cocoa supplementation increased mean blood flow velocity with no effect on cerebrovascular resistance and cerebral vasoreactivity [45]. The acute ingestion of similar cocoa-based supplementation (200 mg to 395 mg total polyphenol) decreased middle cerebral artery velocity and cerebrovascular conductance in response to a computerized cognitive assessment battery with no effect on mean arterial pressure [37]. The remaining study is the only study that investigated the acute and chronic supplementation of polyphenol-rich supplementation (Greek mountain tea) on cerebral hemodynamic registering during completion of a cognitive task, demonstrating that only acute ingestion of either 475 mg (intermediate dose) or 950 mg (high dose) of Greek mountain tea increased oxygenated haemoglobin and oxygen saturation in the prefrontal cortex with total and that deoxygenated hemoglobin only increased using the high dose protocol [28]. From the 7 reviewed studies in Table 3, only 3 studies focused on neuroplasticity and neuroinflammation blood parameters and all of them showed an absence of significant beneficial effects of (poly)phenol-rich supplementation on BDNF or HsCRP compared to placebo [29,31,42]. Two of these three studies also investigated IL-6 and TNF-α as neuroinflammatory biomarkers with one study showing significant increases [42] and the other [29] showing a significant decrease in both parameters in response to resveratrol (200 mg/day) or placebo supplementation.

### 3.7. Meta-Analysis Results

#### 3.7.1. Trail Making Test-A

Data from five trials (comprising 281 participants) were pooled in our meta-analysis [26,40,42,43,44].

Since the studies of Antom et al. [40] and Mastroiacovo et al. [26] each included two doses of polyphenols, results from each condition were considered as an independent study. Pooling the findings yielded a small ES of 0.355 (SE = 0.199, 95% CI −0.035 to 0.746, Z- value = −1.784, *p* = 0.074; Figure 2), with a significant heterogeneity (Q = 16.352, df = 6, *p* = 0.012; I^2^ = 63.308%).

A funnel plot (Figure 3) showed no evidence of publication bias, a conclusion confirmed by Begg and Mazumdar’s rank correlation test (Kendall’s S statistic P-Q = −9.00; tau without continuity correction = 0.429, z = 1.352, *p* = 0.088; tau with continuity correction = 0.381, z = 1.201, *p* = 0.115) and by Egger’s linear regression test (intercept = 0.401, SE = 3.614, 95% CI −8.889 to 54.331, t = 0.111, df = 5, *p* = 0.458). The Duval and Tweedie’s trim-and-fill test did not identify any missing studies.

#### 3.7.2. Trail Making Test-B

Data from six trials (comprising 311 participants) were pooled in our meta-analysis [26,35,40,42,43,44].

Because the studies of Antom et al. [40] and Mastroiacovo et al. [26] each included two doses of polyphenols and because the study of Gleason et al. [35] included 3 intervention periods, results from each of condition were considered as independent studies. Pooling the findings yielded a moderate ES of 0.817 (SE = 0.446, 95% CI −0.058 to 1.692, Z - value=1.831, *p* = 0.092; Figure 4), with a significant heterogeneity (Q = 124.324, df = 5, *p* = 0.000; I^2^ = 95.978%).

Visual inspection of the funnel plot (Figure 5) and the performance of the Begg and Mazumdar’s test (Kendall’s S statistic P-Q = 19.00; tau without continuity correction = 0.422, z = 1.699, *p* = 0.045; tau with continuity correction = 0.400, z = 1.610, *p* = 0.054) provided evidence of publication bias, even though the performance of the Egger’s linear regression test (intercept = − 10.336, SE = 6.171, 95% CI −3.894 to 24.565, t = 1.675, df = 8, *p* = 0.066) did not provide evidence of publication bias. The absence of publication bias was confirmed by the Duval and Tweedie’s trim-and-fill test, which did not identify any missing studies.

#### 3.7.3. Brain-Derived Neurotrophic Factor

Data from three trials comprising 138 participants were pooled in our meta-analysis [29,31,42].

A trivial ES of 0.023 (SE = 0.179, 95% CI −0.329 to 0.374, Z– value = 0.125, *p* = 0.900; Figure 6) was computed, without significant heterogeneity (Q = 1.348, df = 2, *p* = 0.510; I^2^ = 0.000%).

A funnel plot (Figure 7) showed no evidence of publication bias, a conclusion confirmed by Begg and Mazumdar’s rank correlation test (Kendall’s S statistic P-Q = 100; tau without continuity correction = 0.333, z = 0.522, *p* = 0.301; tau with continuity correction = 0.000, z = 0.000, *p* = 0.500) and by Egger’s linear regression test (intercept = 3.847, SE = 2.296, 95% CI −25.225 to 33.020, t = 1.676, df = 1, *p* = 0.171). The Duval and Tweedie’s trim-and-fill test did not identify any missing studies.

#### 3.7.4. High-Sensitivity C-reactive Protein

Data from three trials comprising 138 participants were pooled in our meta-analysis [29,31,42].

A trivial ES of 0.028 (SE = 0.179, 95% CI −0.324 to 0.380, Z – value = −0.156, *p* = 0.876; Figure 8) was computed, without significant heterogeneity (Q = 1.003, df = 2, *p* = 0.606; *I*^2^ = 0.000%).

A funnel plot (Figure 9) showed no evidence of publication bias, a conclusion confirmed by Begg and Mazumdar’s rank correlation test (Kendall’s S statistic P-Q = −1.00; tau without continuity correction = −0.333, z = 0.522, *p* = 0.301; tau with continuity correction = 0.000, z = 0.000, *p* = 0.500) and by Egger’s linear regression test (intercept = −1.465, SE = 3.682, 95% CI −48.260 to 54.331, t = 0.398, df = 1, *p* = 0.379). The Duval and Tweedie’s trim-and-fill test did not identify any missing studies.

#### 3.7.5. Interleukin-6

Data from two trials comprising 106 participants were pooled in our meta-analysis [29,42].

Pooling these findings, a trivial ES of −0.229 (SE = 0.202, 95% CI −0.624 to 0.167, Z – value = −1.134, *p* = 0.257; Figure 10) was computed, without significant heterogeneity (Q = 0.002, df = 1, *p* = 0.966; I^2^ = 0.000%).

#### 3.7.6. Tumor Necrosis Factor alpha

Data from two trials comprising 106 participants were pooled in our meta-analysis [29,42].

Pooling these findings, a trivial ES of −0.257 (SE = 0.202, 95% CI −0.653 to 0.139, Z – value = −1.274, *p* = 0.203; Figure 11) was computed, without significant heterogeneity (Q = 0.324, df = 1, *p* = 0.569; *I*^2^ = 0.000%).

### 3.8. Sensitivity and Cumulative Meta-Analyses

Both sensitivity and cumulative meta-analyses confirmed the reliability and stability of the findings.

## 4. Discussion

To our knowledge, this is the first systematic review and meta-analysis conducted examining the effects of acute and chronic (poly)phenol-rich supplementation on cognitive functions and brain parameters in aging adults. Data regarding changes in a variety of cognitive functions (e.g., psychomotor function, visual processing, attention, executive function, verbal and spatial memory, working memory, and learning abilities) and brain parameters (cerebral blood flow, cerebral hemodynamics, neuroplasticity, and neuroinflammation) following an acute and/or chronic consumption of (poly)phenol-rich supplementation were extracted from the reviewed trials. However, only a few items showed to be sufficiently comparable and were included in the meta-analysis (i.e., executive functions during TMT-A and TMT-B, BDNF, HsCRP, IL-6, and TNF-α). The pooled analysis suggests that chronic administrations (6–26 weeks) of (poly)phenol-rich supplementation have no significant effect on executive functions (TMT-A and TMT-B). In agreement with these findings, some of the included studies have reported that a daily ingestion of 200–300 mg of resveratrol [40,42] as well as 110 mg of isoflavones [43] for a period of 3 to 6 months had no effect on the registered performance during TMT-A and TMT-B tests. However, the ingestion of higher resveratrol doses (1000 mg/day for 3 months) in the study of Antom et al. [40] significantly improved the information processing speed during TMT-A test (i.e., which was not the case using a lower dose of 300 mg/day). Similarly, in response to a daily dose of moderate to high (520–933 mg of total (poly)phenols for 2 months) polyphenol-rich cocoa drinks, Mastroiacovo et al. [26] demonstrated a significant lower time to completion during both TMT-A and TMT-B tests. Discrepancies between findings from the included studies may be linked to the dose (low vs. intermediate vs. high) with intermediate (≈500 mg/day) to high dose (≈1000 mg/day), seemingly improving psychomotor performance of aging adults [26,40]. A dose of 300 mg/day or lower seems only sufficient to stabilize this cognitive performance during the 6–26 weeks intervention period with no more beneficial effect. This suggestion is in line with the findings of Mix and Crews [44], indicating that 180 mg/day of flavonoid-rich Ginkgo biloba extract EGb 761 results in a nonsignificant change in both TMT-A and TMT-B performance compared to baseline. Another included study in the meta-analysis of TMT-B performance [35] revealed that a daily dose of 100 mg soy isoflavones (i.e., low dose) increased performance during TMT-B compared to baseline after one month and significantly decreased after 3- and 6-months of intervention. Taken together, the findings from these studies suggest that a low polyphenol dose (≤300 mg/day) may enhance executive functions up to 1 month of intervention period and may stabilize this performance up to 6–8 weeks of intervention period but do not exert enough effect to counteract the decline in executive functions which can be detected after 3 or 6 months of intervention in aging populations. To counteract the unwanted effect of aging on TMT performances, it seems that at least a daily intermediate to high dose (≥500 mg/day) of polyphenols’ compound should be administered in clinical trials lasting 3 to 6 months. This suggestion extends to future studies investigating the effect of polyphenol-rich supplementation on overall cognitive performance, as Mastroiacovo et al. [26] indicated also a significant increase in “z score” (overall cognitive performance) after 6 months consumption of both intermediate (520 mg/day) and high (933 mg/day) flavanols drinks.

In addition to the used polyphenol dose, the bioavailability of the ingested phenolic compounds is usually reported as an important factor influencing the change on cognition and brain function [57,58]. From a pharmacological perspective, bioavailability is the rate and extent to which the bioactive compound is absorbed and becomes available at the site of action [59]. In other words, this rate reflects the ability of the bioactive compound to cross membranes/barriers and to reach the tissues in an appropriate amount of time to exert its effect [57,60]. Variation in phenolic bioavailability ranges from 0.3% to 43% [21], with isoflavones (≈43%) and Gallic acid (≈38%) representing the most well-absorbed polyphenols, followed by catechins (≈18%), flavanones (≈9%), and quercetin glucosides (≈2.5%); the least well-absorbed polyphenols (<1%) are the proanthocyanidins, the galloylated tea catechins, and the anthocyanins [61,62]. This classification suggests that the most well-absorbed polyphenols such as isoflavones have higher abilities to cross the brain–blood barrier in order to induce a neuroprotective response and to thereby enhance brain and cognitive functions. Findings from the included studies confirm this suggestion for some cognitive functions and indicate that a daily ingestion of 100–110 mg isoflavone for 6 months may improve verbal and visuospatial memory functions and visual-motor function [35,43] while the ingestion of a high dose of anthocyanidins (387 mg/day) only improved the working memory performance during a 2-back cognitive task with no significant effect on other assessed cognitive functions [31]. Therefore, it is recommended that future trials use intermediate to high doses (≥500 mg/day) of polyphenols with an intermediate (9%) to high rate of bioavailability (43%). This strategy would increase the amount of polyphenol metabolites effectively transported/distributed/delivered to target tissues, enhancing the bio-efficacy of the adopted supplementation, thereby impacting the promotion of brain health and the improvement of cognitive performance [59].

Regarding the pooled analysis of the neuroplasticity and neuroinflammation markers, results suggest that the chronic ingestion of 200 mg resveratrol/day or 387 mg anthocyanidins/day have no significant effects on BDNF, HsCRP, IL-6, and TNF-α. All included studies in the analysis of BDNF and Hs-CRP are in line with these findings and showed either an absence of significant changes [29,40] or an increase of a similar magnitude for these biomarkers (BDNF and Hs-CRP) following both (poly)phenols and placebo supplementation [42]. The analyses of IL-6 and TNFα are also in line with the pooled analysis, demonstrating an absence of significant beneficial effects of (poly)phenol-rich supplementation on these biomarkers with either a significant increase [42] or decrease [29] of a similar magnitude following polyphenols and placebo supplementation. However, polyphenols are widely reported to inhibit neuroinflammation by attenuating nitric oxide (NO) production, iNOS induction, NADPH oxidase activation, and subsequent reactive oxygen species (ROS) and proinflammatory cytokine (e.g., IL-1β, IL-6, and TNF-α) generation [20,63,64,65]. The present results of the pooled analysis failed to demonstrate any anti-inflammatory action of chronic anthocyanin [31] and resveratrol [29,42] supplementation on Hs-CRP and/or IL-6 and TNF-α. These results indicate that the employed dose (200 mg to 387 mg) and the bioavailability of the used polyphenol compound (e.g., 0.4 for anthocyanin) of the included studies were not sufficient to exert anti-inflammatory effects. Again, the results from this meta-analysis suggest that at least an intermediate (≥500 mg/day) dose of polyphenols compound with at least intermediate to high bioavailability rates (≥9%) are needed to utilize the bio-efficacy of these food components. This suggestion is in line with previous studies showing that a daily dose of (poly)phenol-rich supplementation with 135 mg to 200 mg total polyphenol has no effect on CRP, IL-6, IL-18, and TNFα [66,67] while blackcurrant juice containing 617 mg total polyphenols has the potential to decrease TNFα, IL-1β, and iNOS mRNA levels while increasing FR-scavenging capacity [68]. Similarly, it has been shown that catechin (estimated bioavailability of 18%) and flavanone (estimated bioavailability of 9%) are highly effective in inhibiting TNF-α release and inflammatory signaling in glial cells and in protecting against neuroinflammatory injury [69], while quercetin supplementation with a lower estimated bioavailability rate (≈2.5%) fails to demonstrate any anti-inflammatory action [70].

The impact of using (poly)phenol compounds at efficient doses and bioavailability rates to increase its efficacy have also been confirmed by the majority of studies in Table 3. Regarding the effect of rich-(poly)phenol supplementation on CBF, Marsh et al. [37] showed that flavanol-rich cocoa containing 200 mg to 395 mg total polyphenols failed to improve CBF responses while a higher dose of 494 mg [30] and 900 mg [45] showed increases in mean CBF velocity, neuronal activities, and regional CBF in the anterior cingulate and central opercular cortex. Generally, the beneficial effects of (poly)phenols on CBF and neuronal activity were reported to be due to their positive impact on several measures of endothelial function and other aspects of the vasculature through an activation of the NO system [25,71,72,73]. Particularly, polyphenol compounds were hypothesized to promote nitric oxide (NO) synthesis (an important contributor to flow-mediated dilation [74]) by enhancing nitric oxide synthase (NOS) activity and NO bioavailability through limiting NO scavenging by ROS [75]. To exert these beneficial effects on the brain, a sufficient amount of polyphenol metabolites would be expected to cross the blood–brain barrier [76] towards their specific binding sites on neurons, triggering the activation of various downstream kinases (e.g., mitogen-activated protein (MAP) and PI3 kinase pathways), thus inducing the opening of agonist-activated ion channels and leading to increased neuronal activity accounting for increased CBF [77,78]. The abovementioned mechanism indicates that an insufficient amount of polyphenol metabolites, beyond the brain–blood barrier, would blind the bio-efficacy of (poly)phenol-rich supplementation on brain health. Such a mechanism explains the divergent effects of flavanol rich cocoa on CBF (absence of beneficial effect using 200–395 mg vs. improvement using ≥495 mg of polyphenol dose) and, most importantly, suggests that an efficient (poly)phenol dose (≥500 mg) containing phenolic compound with moderate to high bioavailability (≥9%) is indispensable to improve brain health in older adults.

*Strengths and Weaknesses*: This is the first systematic review and meta-analysis evaluating the effect of acute and/or chronic (poly)phenols-rich supplementation on cognitive and brain function in healthy aging adults. Strengths of the study include a comprehensive coverage of the current literature via the utilization of a wide range of key words related to cognition and brain, the searching through two scholarly databases (as recommended by the Cochrane Association guidelines and good practices for conducting systematic reviews), the absence of language restriction, and a careful appraisal of study quality. However, given some methodological issues, results must be interpreted with caution. These methodological issues include (i) the application of some broad key words (e.g., “injury”, “patient”, “disease”, and “impairment”) that were filtered out of the systematic search (modifier: NOT) and, therefore, some relevant results would have been likely filtered, and (ii) the relatively medium sample sizes of the individual studies which used a large variety of cognitive task batteries, imaging technics, neuroprotective markers, observation periods, and supplementation doses. Therefore, further rigorous studies on this issue, including more than two scholarly databases (e.g., Web of Science, Pubmed, Embase, and Cochrane), are warranted.

## 5. Conclusions

The evaluation of the beneficial impact of (poly)phenols on cognitive and brain function during aging has recently garnered increased interest as food industries are continually involved in developing new attractive alternative therapeutic agents. This meta-analysis failed to provide evidence regarding the neuroprotective and anti-inflammatory effect of (poly)phenol supplementation in aging adults. However, findings from individual studies included in this systematic review suggest that polyphenol-rich supplementation may improve some cognitive and brain functions in older adults. The beneficial effects of (poly)phenols on the studied brain health parameters appear to depend on both ingested dose and bioavailability, which vary greatly. Particularly, the present systematic review and meta-analysis suggest at least an intermediate dose (≥500 mg) of polyphenols with intermediate (≈9%) to high (43%) bioavailability rate/profiles (e.g., isoflavones, gallic acid, catechin, and flavanones) is needed to cross the blood–brain barrier and to exert a healthy effect. These findings provide better general insight into (poly)phenols effect on cognitive and brain function and present further information, including appropriate dosage and bioavailability profiles of the consumed compounds, to provide health benefits in older brain structures and functions. This information should be useful for the design and interpretation of intervention studies investigating the health effects of (poly)phenols. However, as the number of available studies concerning the described topic was rather small, it is essential to conduct further research with adjusted supplementation and measuring methodology to create improved approaches to take advantage of health promoting effects of these compounds in older adults.

## Figures and Tables

**Figure 1 jcm-09-00835-f001:**
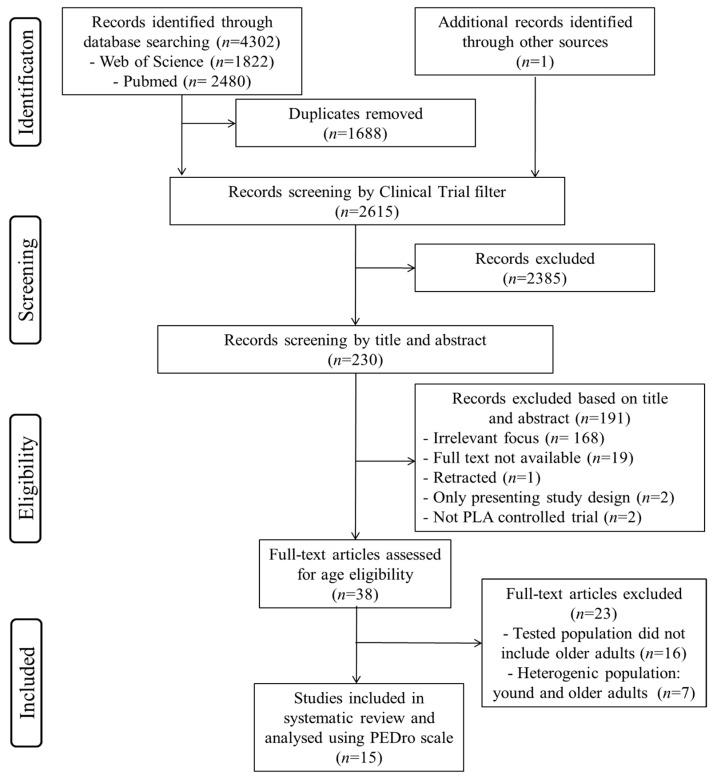
Flow diagram of the literature selection process.

**Figure 2 jcm-09-00835-f002:**
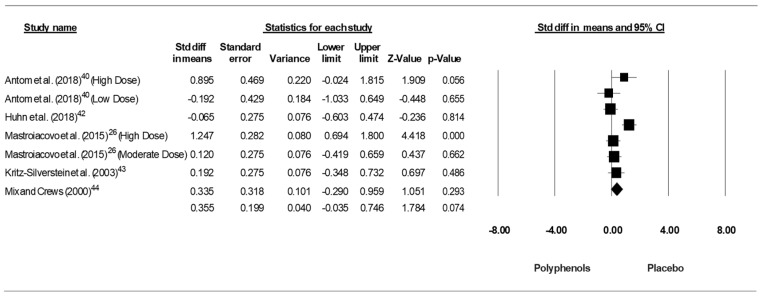
Forest plot of studies investigating the effect of (poly)phenols-rich supplementation on Trail Makin Test (TMT-A).

**Figure 3 jcm-09-00835-f003:**
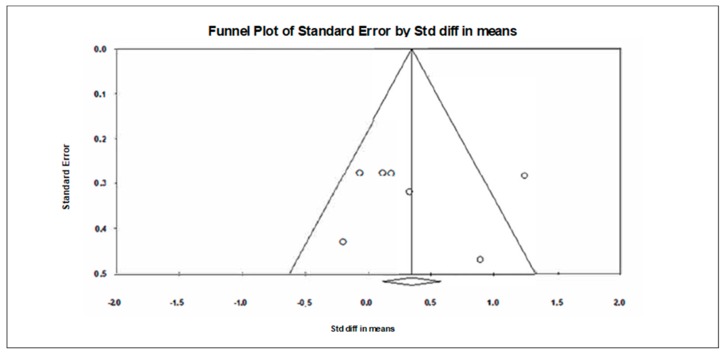
Funnel plot for executive functions in TMT-A showing no evidence of publication bias.

**Figure 4 jcm-09-00835-f004:**
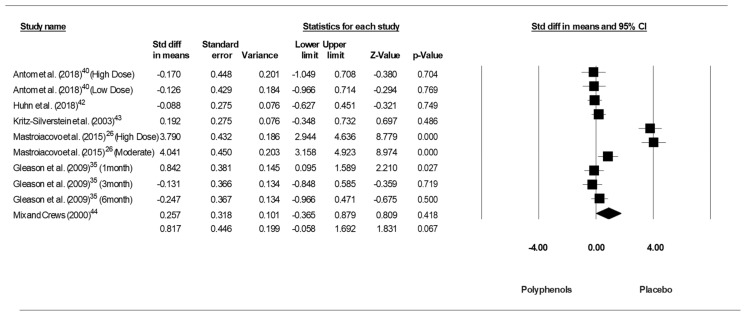
Forest plot of studies investigating the effect of (poly)phenols-rich supplementation on Trail Making Test (TMT-B).

**Figure 5 jcm-09-00835-f005:**
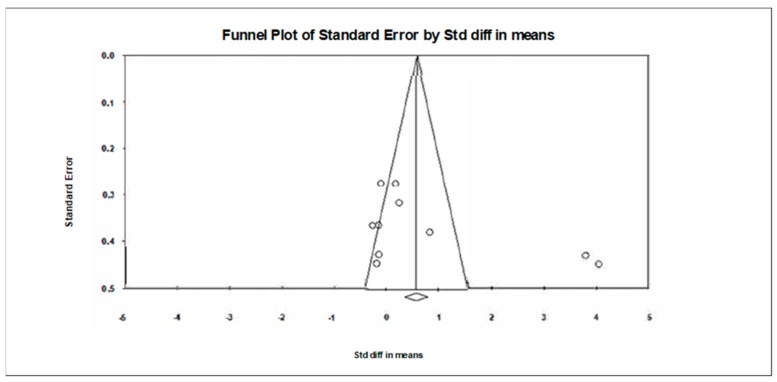
Funnel plot for executive functions in TMT-B showing evidence of publication bias.

**Figure 6 jcm-09-00835-f006:**
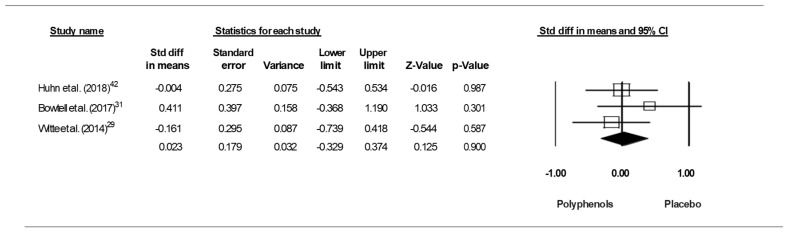
Forest plot of studies investigating the effect of (poly)phenols-rich supplementation on brain-derived neurotrophic factor.

**Figure 7 jcm-09-00835-f007:**
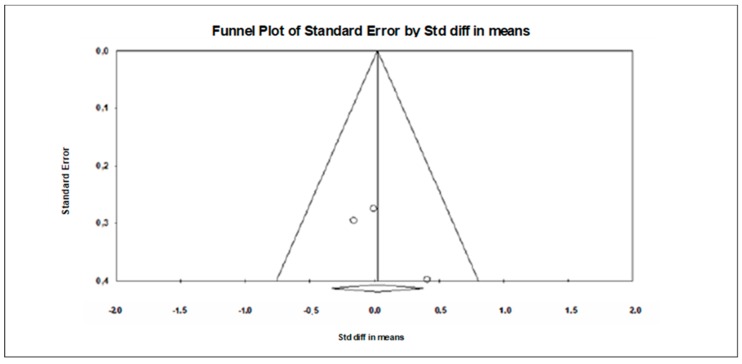
Funnel plot for brain-derived neurotrophic factor showing no evidence of publication bias.

**Figure 8 jcm-09-00835-f008:**
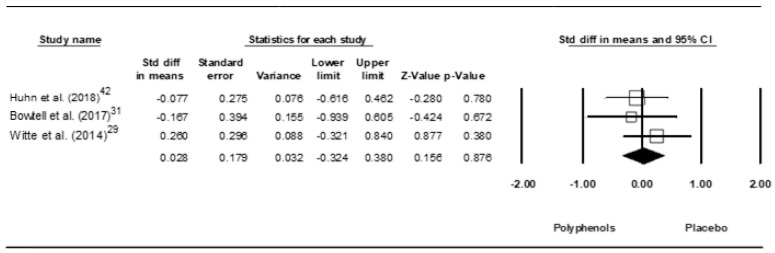
Forest plot of studies investigating the effect of (poly)phenols-rich supplementation on high-sensitivity C-reactive protein.

**Figure 9 jcm-09-00835-f009:**
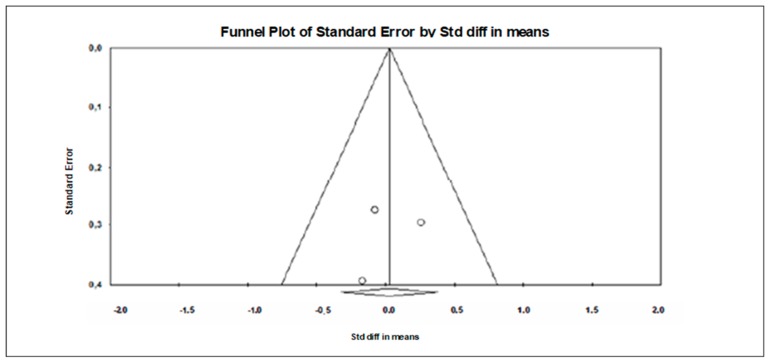
Funnel plot for high-sensitivity C-reactive protein showing no evidence of publication bias.

**Figure 10 jcm-09-00835-f010:**
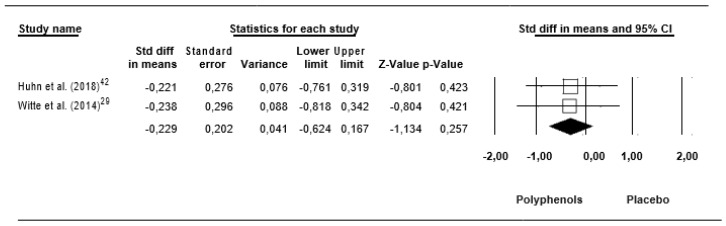
Forest plot of studies investigating the effect of (poly)phenols-rich supplementation on interleukin 6.

**Figure 11 jcm-09-00835-f011:**
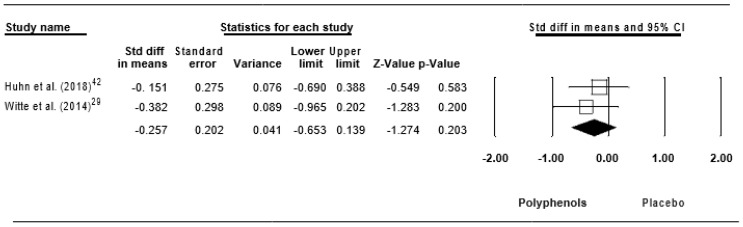
Forest plot of studies investigating the effect of (poly)phenols-rich supplementation on tumor necrosis factor alpha.

**Table 1 jcm-09-00835-t001:** PICOS (population, intervention, comparison, outcome, and study design) criteria for inclusion of studies.

Parameter	Inclusion Criterion
Participants	Aging adults (55 years old and over)
Intervention	Acute and/or chronic (poly)phenols-rich supplementation
Comparison	Any
Outcome	Cognitive functions (e.g., overall cognition, psychomotor performance, executive function, processing speed, attention, language, verbal memory, and visual memory) and neuroprotective measures (e.g., brain perfusion, brain activity, cerebral hemodynamics, cerebral blood flow (CBF), neuroplasticity, and neuroinflammation)
Study design	Randomized controlled trial

**Table 2 jcm-09-00835-t002:** Effect of polyphenols-rich supplementation on cognitive functions.

Authors	Study Design	Treatment	Phenolyc Content	Dose	Duration	Washout	Study Population	Effect on Cognitive Functions	Assessing Methods (e.g., Cognitive Battery Test)
**Antom et al. (2018) [40]**	Double-blind, phase II, randomized, placebo-controlled pilot study	Resveratrol	Not mentioned	Low dose: 300 mg/day; high dose: 1000 mg/day	Chronic: 90 Days	N/A	*n* = 32 (10 PLA, 12 low dose, 10 high dose); sedentary older adults, mean age: 73.34 ± 7.02 years old (65–93 years)	↑ significantly improves psychomotor speed on the TMT-A using 1000 mg/day; ↔ nonsignificant effect on the other cognitive functions (visual attention, working memory, verbal fluency, and semantic memory)	Trail Making Test (TMT) assessing executive functions: TMT-A = information processing speed, digits forward and backward (auditory attention: forward) and working memory (backward), digit symbol substitution test (attention and processing speed), Erikson–Flanker task (executive function by measuring response inhibition), controlled oral word association (verbal fluency), Hopkins Verbal Learning Test-Revised: HVLT (verbal learning, recognition memory, and recall), and task switching (accuracy and reaction time)
**Bensalem et al. (2019) [41]**	Bicentric, randomized, double-blind, placebo-controlled trial	polyphenol-rich extract from grape and blueberry (PEGB)	258 mg flavonoids/day	600 mg/day: 2 capsules (each one contains 300 mg PEGB)/day	chronic: 24 weeks	N/A	*n* = 190 (92 PEGB group, 98 PLA group), healthy elderly subjects, mean age: 64.66 ± 2.91 years old (60–70 years)	↑ significantly improves VRM-free recall; ↔ nonsignificant effect on the other cognitive functions (PAL, VRMR, SSP, and reverse SSP)	Cambridge Neuropsychological Test Automated Battery (CANTAB): Paired Associate Learning (PAL) test (visuospatial learning and episodic memory); verbal recall memory: VRM (episodic RM using VRM-free recall (VRMFR) and VR recognition memory using (VRMR) and working memory using the Special Span (SSP) and the reverse SSP tests
**Bowtell et al. (2017) [31]**	Randomized, double-blind, placebo-controlled parallel trial	Anthocyanin-rich blueberry concentrate	387 mg anthocyanidins/day	30 mL blueberry concentrate (diluted to 240 mL total volume with tap water); 387 mg anthocyanidins/day	chronic: 12 weeks	N/A	*n* = 26 (12 blueberry, 14 PLA), older adults, mean age: blueberry group = 67.5 ± 0.9, PLA group = 69 ± 0.9	↑ significantly improves working memory performance during 2 back cognitive task; nonsignificant improvement during 1 back cognitive task; ↔ nonsignificant effect on the other cognitive functions	The cognitive battery of tests (CogStateLtd.): detection task (psychomotor function), the Groton maze timed chase test (speed of visual processing), the Groton maze learning test with a delayed recall component (executive function and delayed record), identification task (attention), international shopping list task with delayed recall (verbal learning and delayed recall), 1-back and 2-back memory tasks (working memory). The speed and accuracy of responses were quantified.
**Gleason et al. (2009) [35]**	Double-blind, randomized, placebo-controlled, parallel-group pilot study	Purified glycosidic isoflavones	100 mg isoflavones	100 mg soy-isoflavones/day	chronic: 6 months	N/A	*n* = 30, (15 isoflavones, 15 PLA), older adults: mean age: isoflavone group = 73 ± 7.9, PLA group: 74.3 ± 6.3, (range: 62–89 years)	↑ significantly improves cognitive performances on 2 tests of verbal and visuospatial memory (Rey complex figure and visual spatial learning tests), verbal/language fluency test, and visual-motor function tests; ↔ nonsignificant effect on two tests of verbal learning and recall (Buschke Selective Reminding Test and Paragraph Recall Test), language (Boston Naming test), one test of executive function (mazes); ↓ significantly decreases performance during two tests of executive functions (Trail Making Test part B TMT-B and Stroop Color-Word test)	Battery of neuropsychological measures: verbal and visuospatial memory (Buschke Selective Reminding test, Paragraph Recall, Rey Complex Figure test, Visual Spatial Learning test); language (Boston Naming test); language fluency (FAS, animal fluency); visual-motor function (Rey Complex Figure test copy, Grooved Pegboard), and executive function (Stroop Color Word test, Mazes, TMT-B = cognitive flexibility)
**Howes et al. (2004) [36]**	Double blind, randomized, placebo-controlled, counterbalanced trial	Isoflavone-rich extract from red clover	25 mg of formononetin, 2.5 mg of biochanin and <1 mg of genistein and daidzein	Two tablets/day (~55 mg isoflavone/day)	chronic: 6 months	1 month	*n* = 30, (15 isoflavones, 15 PLA), older women, mean age: isoflavones group = 68.5 ± 6.6 years; placebo group = 67.7 ± 5.5 years	↑ significantly improves cognitive performance during block design test (a test of visual-spatial intelligence); ↔ nonsignificant effect on the other tests; ↓ significantly decreases performances during digit recall and verbal memory 2 tests (all not significant if correction to multiple comparisons is made)	Tests of speed of information processing (trail A and digit symbol); tests of memory (memory 1 and 2, verbal memory 1 and 2, and visual memory 1 and 2); tests of verbal ability (Boston naming test, FAS test, animal naming test, and similarities naming test); tests of frontal cortex function (arithmetic test, trail B test, and block design test); digit recall
**Huhn et al. (2018) [42]**	Double-blind, randomized controlled trial	Resveratrol	200 mg resveratrol/day	Two pills of 100 mg resveratrol per day (total daily dose/day = 200 mg/day)	chronic: 26 weeks	N/A	*n* = 60 (30 resveratrol group, 30 PLA), elderly participants, range age: 60–79 years	nonsignificant improvement on pattern recognition memory; ↔ nonsignificant effect on verbal memory performance	Verbal memory performance (learning ability, delayed recall, rate of forgetting, and recognition) were assessed using the German version of the California Verbal Learning Task (CVLT); attention and mental flexibility were assessed using the TMT-A and TMT-B. Pattern recognition performance was assessed with the ModBent task
**Kritz-Silverstein et al. (2003) [43]**	Double-blind, randomized, placebo-controlled trial	Soy-extracted isoflavones	110 mg total isoflavones/day	Two pills of 55 mg of soy-extracted isoflavones per day (total daily dose = 110 mg/day)	chronic: 6 months	N/A	*n* = 53 (27 treatment, 26 PLA), older women, mean age: treatment group = 60 ± 4, PLA: 62 ± 6	↑ significantly improves category fluency verbal memory performance; nonsignificant improvement on trail B and the immediate and delayed logical memory and recall test; ↔ nonsignificant effect on trail A	Cognitive function tests: trails A and B (assess visuomotor tracking and attention), category fluency (assess verbal memory), and logical memory and recall tests (a paragraph recall test assessing immediate and delayed verbal memory).
**Marsh et al. (2017) [37]**	Counterbalanced within-subject crossover design	Chocolate with a high concentration of polyphenols-rich cocoa	Total polyphenols: ≈85 mg: white chocolate, 200 mg: milk chocolate, 395 mg: Dark chocolate	84 g dark chocolate (80% cacao), 87 g milk chocolate (35% cacao), or 85 g white chocolate (0% cacao) per daily	Acute	1 week	*n* = 12 (counterbalanced design), older women, mean age: 57.3 ± 5.3 y)	↔ nonsignificant effect on any of the assessed cognitive functions	7 CogState measures: verbal memory, psychomotor memory, visual attention, working memory (one back), working memory (two back), visual memory, and verbal recall memory
**Mastroiacovo et al. (2015) [26]**	Double-blind, controlled, parallel-arm study	Cocoa flavanol drinks (high, moderate, and low flavanol contents)	Total flavanols: high flavanol drink (993 mg), moderate flavanols drink (520 mg), low flavanols drink (48 mg)	250 mL drink with high, moderate, or low flavanol content/day	chronic: 8 weeks	N/A	*n* = 90 (30 for each study’s arm), older adults, age > 60 years old	↑ significant improvement in performance of TMT-A and TMT-B and overall cognitive performance “z score” after consumption of the high and the intermediate flavanol drinks; ↑significant improvement in the Verbal Fluency Test (VFT) score using the high flavanols drink; ↔ nonsignificant effect on Mini-Mental State Examination (MMSE)	Mini-Mental State Examination (MMSE); TMT-A and TMT-B; the Verbal Fluency Test (VFT); and overall cognitive function (z score)
**Mix & Crews. (2000) [44]**	Double-blind, fixed-dose, placebo-controlled, parallel-group design	Ginkgo biloba extract EGb 761	Not mentioned	180 mg/day	6 weeks	N/A	*n* = 48 (*n* of each arm: not mentioned), older adults, age range: 55–86 years old	↑ significant improvement on one task assessing speed of processing abilities (i.e., color-naming task of the Stroop Color and Word Test); nonsignificant improvement in the majorities of the remaining tasks that involved a timed, speed of processing component (e.g., trail making test A and B); ↔ nonsignificant effect on the four objective memory measures (i.e., logical memory I and II and visual reproduction I and II)	Stroop Color and Word Test, TMT-A and TMT-B, Wechsler Memory Scale—Revised (WMS-R), Logical Memory I and II (LM I and II), and Visual Reproduction I and II subtests (VR I and II)
**Whyte et al. (2018) [27]**	Randomized, double blinded, placebo-controlled trial	1000 mg capsules of wild blueberry (WB): WBP500WBP1000 and WBE111 (purified WB extract)	Total polyohenol contents: WBP500 (35 mg/capsule) powder); WBP1000 (70 mg/capsule); WBE111 (50 mg/capsule)	1 daily dose of either WBP500, WBP1000, or WBE111	Chronic intervention (3 and 6 months)	N/A	*n* = 122 (30 PLA, 30 WBP500, 31 WBP1000, and 31 WBE111), older adults, mean age: 71 ± 4 years old (65–80 years)	↑ significant improvement of verbal and short-term spatial episodic memory performances with better delayed word recognition during the RAVLT and better recall of sequences during the Corsi Block task following WBE111 at 3 months compared to PLA, ↔ nonsignificant effect on working memory and executive function at 3 months follow-up ↔ nonsignificant effect for all cognitive performance at 6 months follow-up	Battery of cognitive tasks targeting episodic memory (verbal episodic memory using the Rey’s Auditory Verbal Learning task (RAVLT), visual episodic memory using an object recognition task, and short-term spatial episodic memory using the Corsi Blocks task), working memory (using serial subtractions and Sternberg memory scanning tasks), and executive function (using the Modified Attention Network Task (MANT) and Stroop task)
**Wightman et al. (2018) [28]**	Double blind, randomized, placebo controlled, parallel groups study	Polyphenol-rich Greek mountain tea (Sideritis scardica)	Potal phenolic content = 6.25% of the 20% Greek mountain tee extract	475 or 950 mg of Greek mountain tea daily	Acute and chrnic (4 weeks)	N/A	*n* = 140 (*n* of each arm: not mentioned), older adults, mean age 60.3 years old	Acute and chronic effects of the Greek mountain tea with ↑ significant improvement in working memory (fewer false alarm during RVIP test) and higher episodic memory accuracy (during the picture recognition task) using the higher (950 mg) dose; ↑ significant improvement in speed of attention (derived from reaction time during numerical working memory, choice reaction time, and RVIP tasks) using both intermediate (475 mg) and high doses (950 mg) compared to active Ginko control (240 mg)	Cognitive function tests battery (www.cognitivetesting.co.uk) assessing accuracy and speed of attention (choice reaction time test), working memory (numerical working memory task, Serial 3s and 7s tasks, and Rapid Visual Information Processing (RVIP) task), and episodic memory (delayed word recall, delayed name/face recall, delayed picture recognition, and delayed word recognition tasks) performances
**Witte et al. (2014) [29]**	Double blind, randomized, placebo controlled, parallel groups study	Resveratrol	Not mentioned	200 mg/d	26 weeks	N/A	*n* = 46 (23 resveratrol, 23 PLA), older female, mean age: resveratrol group = 65 ± 7 years old, PLA = 64 ± 5 years old	↑ significant improvement on retention ability; nonsignificant improvement on delayed recall and recognition, ↔ nonsignificant effect on learning ability using AVLT	Memory performance (i.e., retention, delayed recall, and recognition) and learning ability were assessed using the Auditory Verbal Learning Test (AVLT)

Placebo (PLA), Trail Making Test (TMT), Cambridge Neuropsychological Test Automated Battery (CANTAB), Hopkins Verbal Learning Test-Revised (HVLT), Paired Associate Learning (PAL), Verbal Recall Memory (VRM), Verbal Recall Memory Free Recall (VRMFR), Special Span (SSP), Mini-Mental State Examination (MMSE), Verbal Fluency Test (VFT), Wechsler Memory Scale—Revised (WMS-R), Visual Reproduction (VR), Rey’s Auditory Verbal Learning task (RAVLT), Rapid Visual Information Processing (RVIP), Auditory Verbal Learning Test (AVLT), Modified Attention Network Task (MANT), Wild Blueberry (WB).

**Table 3 jcm-09-00835-t003:** Effect of (poly)phenols-rich supplementation on neuroprotective measures.

Authors	Study Design	Treatment	Phenolyc Content	Dose	Duration	Washout	Study Population	Effect on Neuroprotective Measures	Assessing Methods (e.g., Cognitive Battery Test)
**Bowtell et al. (2017) [31]**	Randomized, double-blind, placebo-controlled parallel trial	Anthocyanin-rich blueberry concentrate	387 mg anthocyanidins/day	30 mL blueberry concentrate (diluted to 240 mL total volume with tap water); 387 mg anthocyanidins/day	chronic: 12 weeks	N/A	*n* = 26 (12 blueberry, 14 PLA), older adults, mean age: blueberry group = 67.5 ± 0.9, PLA group = 69 ± 0.9	↑ significantly improves task-related brain activation and increases resting regional grey matter perfusion in the parietal and occipital lobes; ↔ nonsignificant effect on BDNF and hs-CRP	1.5 T MRI scanner during numerical Stroop test to quantify task-related activation; Arterial Spin Labelling Magnetic Resonance Imaging (ASL MRI) technique to determine quantitative resting brain perfusion; blood parameters (BDNF, hs-CRP)
**Huhn et al. (2018) [42]**	Double-blind, randomized controlled trial	Resveratrol	200 mg resveratrol/day	Two pills of 100 mg resveratrol per day (total daily dose/day = 200 mg/day)	chronic: 26 weeks	N/A	*n* = 60 (30 resveratrol group, 30 PLA), elderly participants, range age: 60–79 years	↔ nonsignificant effect on hippocampus subfield volumes, mean weighted image diffusivity, and hippocampus connectivity; ↔ nonsignificant effect on BDNF, hs-CRP, TNF-α, and IL-6	Anatomical MRI for hippocampal volumetry was acquired at a Siemens Magnetom 7 Tesla system; blood parameters (BDNF, hs-CRP, IL-6, and TNF-α)
**Lamport et al. (2015) [30]**	Randomized, counterbalanced double-blind, crossover trial	Cocoa flavanols	High flavanol drink (494 mg), low flavanols drink (29 mg)	330 mL containing high or low flavanols content daily	Acute	1 week	*n* = 18 (counterbalanced) older adults, mean age: 61 ± 5 years old (55–65 years)	↑ significantly increases regional CBF in the anterior cingulate cortex and central opercular cortex	Arterial Spin Labelling Functional Magnetic Resonance Imaging (ASL fMRI) to assess resting regional perfusion
**Marsh et al. (2017) [37]**	Counterbalanced within-subject crossover design	Chocolate with a high concentration of polyphenols-rich cocoa	Total polyphenols: ≈85 mg: white chocolate, 200 mg: milk chocolate, 395 mg: dark chocolate	84 g dark chocolate (80% cacao), 87 g milk chocolate (35% cacao), or 85 g white chocolate (0% cacao) per daily	Acute	1 week	*n* = 12 (counterbalanced design), older women, mean age: 57.3 ± 5.3 y)	↓ significantly decreases CBF responses (i.e., middle cerebral artery velocity and cerebrovascular conductance) during the cognitive tasks using milk and dark chocolate; ↔ nonsignificant effect on mean arterial pressure	Transcranial Doppler (TCD) (Spencer Technologies) to assess cerebral blood flow velocity (CBFv) responses to a computerized cognitive assessment battery (CogState)
**Sorond et al. (2008) [45]**	Randomized, double-blind, parallel arm trial	Cocoa flavanol drink (flavanol-rich cocoa (FRC) and flavanol-poor cocoa (FPC))	450 mg flavanol cocoa in each 450 mg FRC packet drink and 18.2 mg flavanol cocoa in each 450 mg FPC drink packet	2 packets daily (900 mg/day)	1 and 2 weeks	N/A	*n* = 21 (*n* of each arm: not mentioned), healthy elderly volunteers, mean age = 72 ± 6 years old, (59–83 years)	↑ significantly increases mean Blood Flow Velocity (MFV) with 8% ± 4% during the first week and 10% ± 4% during the two weeks, ↔ nonsignificant effect on cerebrovascular resistance (CVR) and cerebral vasoreactivity (VR)	Transcranial Doppler (TCD) ultrasonography; outcome: Mean Blood Flow Velocity (MBFV), cerebrovascular resistance (CVR), and cerebral vasoreactivity (VR) in the middle cerebral artery (MCA)
**Wightman et al. (2018) [28]**	Double Blind, randomized, placebo controlled, parallel groups study	Polyphenol-rich Greek mountain tea (Sideritis scardica)	Total phenolic content = 6.25% of the 20% Greek mountain tee extract	475 or 950 mg of Greek mountain tea daily	Acute and chronic (4 weeks)	N/A	*n* = 57 (*n* of each arm: not mentioned), older adults, mean age 60.3 years old	The acute ingestion of the Grek mountain tea during completion of cognitively demanding tasks: ↑ significantly improves oxygenated haemoglobin (HbO) and oxygen saturation (Ox%) in the prefrontal cortex using both intermediate (475 mg) and high dose (950 mg) compared to active Ginko control (240 mg), and ↑ significantly improves total (THb) and deoxygenated (Hb) haemoglobin only using the high dose (950 mg) compared to active Ginko control (240 mg); ↔ no significant effect of the chronic ingestion at day 28	Near-Infrared Spectroscopy (NIRS) during completion of cognitive task to assess cerebral hemodynamics/blood flow including total hemoglobin (total-Hb), oxygenated hemoglobin (oxy-Hb), deoxygenated hemoglobin (deoxy-Hb), and oxygen saturation (Ox%) in the prefrontal cortex
**Witte et al. (2014) [29]**	Double blind, randomized, placebo controlled, parallel groups study	Resveratrol	Not mentioned	200 mg/day	26 weeks	N/A	*n* = 46 (23 resveratrol, 23 PLA), older female, mean age: resveratrol group = 65 ± 7 years old, PLA = 64 ± 5 years old	↑significantly improves hippocampal functional connectivity (FC); ↔ nonsignificant effect on total gray matter volume or in the volume or microstructure of the hippocampus, ↔ nonsignificant effect on BDNF, Hs-CRP, and IL-6.	Neuroimaging (MRI 3 tesla) to assess volume, microstructure, and functional connectivity (FC) of the hippocampus; blood parameters (BDNF, hs-CRP, IL-6, and TNF-α)

Placebo (PLA), Magnetic Resonance Imaging (MRI), Arterial Spin Labelling Magnetic Resonance Imaging (ASL MRI), Brain-derived neurotrophic factor (BDNF), High-sensitivity C-reactive Protein (hs-CRP), Interleukin-6 (IL-6), Tumor Necrosis Factor alpha (TNF-α), Arterial Spin Labelling Functional Magnetic Resonance Imaging (ASL fMRI), Transcranial Doppler (TCD), Cerebral Blood Flow velocity (CBFv), Mean Blood Flow Velocity (MBFV), Cerebrovascular Resistance (CVR), Cerebral Vasoreactivity (VR), Middle Cerebral Artery (MCA), Near-Infrared Spectroscopy (NIRS), Total hemoglobin (total-Hb), Oxygenated hemoglobin (oxy-Hb), Deoxygenated hemoglobin (deoxy-Hb) and oxygen saturation (Ox%), Functional Connectivity (FC).

**Table 4 jcm-09-00835-t004:** Methodological quality of the studies with (poly)phenols-rich supplementation assessed with the PEDro scale.

	Items	Antom et al. (2018) [40]	Bensalem et al. (2019) [41]	Bowtell et al. (2017) [31]	Gleason et al. (2009) [35]	Howes et al. (2004) [36]	Huhn et al. (2018) [42]	Kritz-Silverstein et al. (2003) [43]	Lamport et al. (2015) [30]	Marsh et al. (2017) [37]	Mastroiacovo et al. (2015) [26]	Mix and Crews (2000) [44]	Sorond et al. (2008) [45]	Whyte et al. (2018) [27]	Wightman et al. (2018) [28]	Witte et al. (2014) [29]
11	Eligibility criteria were specified.	+	+	+	+	+	+	+	+	+	+	+	+	+	+	+
22	Subjects were randomly allocated to groups (in a crossover study, subjects were randomly allocated an order in which treatments were received).	+	+	+	+	+	+	+	+	+	+	+	+	+	+	+
33	Allocation was concealed.	-	+	+	+	+	+	+	+	-	+	+	+	+	+	+
44	The groups were similar at baseline regarding the most important prognostic indicators.	+	+	+	+	+	+	+	+	+	+	+	+	+	+	+
55	There was blinding of all subjects.	+	+	+	+	+	+	+	+	+	+	+	+	+	+	+
66	There was blinding of all therapists who administered the therapy.	+	+	+	+	+	+	+	+	-	+	+	+	+	+	+
77	There was blinding of all assessors who measured at least one key outcome.	-	-	-	-	-	-	-	-	-	-	-	-	-	-	-
88	Measures of at least one key outcome were obtained from more than 85% of the subjects initially allocated to groups.	+	+	+	+	+	+	+	+	+	+	+	+	+	+	+
99	All subjects for whom outcome measures were available received the treatment or control condition as allocated or, where this was not the case, data for at least one key outcome was analysed by “intention to treat”.	+	+	+	+	+	+	+	+	+	+	+	+	+	+	+
110	The results of between-group statistical comparisons are reported for at least one key outcome.	+	+	+	+	+	+	+	+	+	+	+	+	+	+	+
111	The study provides both point measures and measures of variability for at least one key outcome.	+	+	+	+	+	+	+	+	+	+	+	+	+	+	+
	Total score	8	9	9	9	9	9	9	9	7	9	9	9	9	9	9

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
