# Peer review of "The Effect of (Poly)phenol-Rich Interventions on Cognitive Functions and Neuroprotective Measures in Healthy Aging Adults: A Systematic Review and Meta-Analysis"

_jcm, 2020, doi:10.3390/jcm9030835_

Round 1
Reviewer 1 Report
In this study Ammar et al. performed a systematic review and meta-analysis on the effecs of a polyphenol-rich diet on cognition and potentially neuroprotective changes in cerebral parameters.
The article follows the PRISMA guidelines and is well-structured. The provided data are presented and analysed (both narratively and statistically) in a clear and concise manner.
Minor comments/revisions:
- The authors state that an elderly population was chosen as the intended participant group, but no specific cut-off age is provided from which age downward studies were excluded based on the age of their participants.
- While the search strategy is included as an overview in the appendix, the separate Pubmed and Web of Science search strategies should be provided in detail (modifiers, which key words were connected to MeSH terms, etc.). Furthermore, it should be clarified whether the search was applied to titles only, or title and abstract, etc.
- The authors have utilized a wide range of key words regarding their intended outcome parameters (cognition and cerebral parameters), which would have significantly enlarged the number of results yielded by the systematic search. On the other hand, iit seems to me that at least some of the applied key words that were filtered out of the systematic search (modifier: NOT) are so broad - e.g. 'injury',, 'patient', 'disease', 'impairment' - that it is likely that some relevant results would have been filtered out because of this. An explanation of this choice in 'Limitations of the study' (see below) would benefit the text.
- While the table of the PICOS criteria (Appendix B) lists p.27 for a discussion of the limitations of this study, I don't think there's a real discussion of this point in the text. A small paragraph in the 'Discussion' section should be added which is dedicated to this aspect (for example the different parameters utilized by the individual studies, as well as differing lengths of observation periods, dosages; why only Web of Science and Pubmed were searched (Embase or Cochrane might have been useful additions); etc.).
- There are some small language issues which, when corrected, will make the manuscript more easily readable. Throughout the text there are some places in which the wrong verb conjugations have been used (do/does, etc.) and some nouns lack their respective articles before them
Within tables 2 and 3, the frequently used phrase 'significantly improve/decrease/increase' should be corrected to the right tense ("improves") or changed from verb to noun (significant improvement, etc.) as used at other places within the same tables.
Author Response
Reviewer #1:
In this study Ammar et al. performed a systematic review and meta-analysis on the effecs of a polyphenol-rich diet on cognition and potentially neuroprotective changes in cerebral parameters.
The article follows the PRISMA guidelines and is well-structured. The provided data are presented and analysed (both narratively and statistically) in a clear and concise manner.
The authors would like to thank the reviewer for the insightful and constructive comments on our work. We have carefully considered all of the suggestions and have revised the manuscript accordingly. We believe that our manuscript is much stronger as a result of making these modifications.
Please find below the authors’ responses to the individual comments.
Minor comments/revisions:
The authors state that an elderly population was chosen as the intended participant group, but no specific cut-off age is provided from which age downward studies were excluded based on the age of their participants.
Response
Thank you for your comment, a cut-off age has been added in the inclusion and exclusion criteria and in PICOS table.
Adults aged 55 years old and over were considered as aging population (https://www.ncbi.nlm.nih.gov/pubmed/11815703)
While the search strategy is included as an overview in the appendix, the separate Pubmed and Web of Science search strategies should be provided in detail (modifiers, which key words were connected to MeSH terms, etc.). Furthermore, it should be clarified whether the search was applied to titles only, or title and abstract, etc.
Response
Thank you for your comment, the exact Pubmed and Web of Science search strategies have been added in the appendix S1 and a sentence clarifying that the search was applied to titles and abstract has been added in the method section.
The authors have utilized a wide range of key words regarding their intended outcome parameters (cognition and cerebral parameters), which would have significantly enlarged the number of results yielded by the systematic search. On the other hand, iit seems to me that at least some of the applied key words that were filtered out of the systematic search (modifier: NOT) are so broad - e.g. 'injury',, 'patient', 'disease', 'impairment' - that it is likely that some relevant results would have been filtered out because of this. An explanation of this choice in 'Limitations of the study' (see below) would benefit the text.
While the table of the PICOS criteria (Appendix B) lists p.27 for a discussion of the limitations of this study, I don't think there's a real discussion of this point in the text. A small paragraph in the 'Discussion' section should be added which is dedicated to this aspect (for example the different parameters utilized by the individual studies, as well as differing lengths of observation periods, dosages; why only Web of Science and Pubmed were searched (Embase or Cochrane might have been useful additions); etc.).
Thank you for raising all these points.
A “Strength and Weakness” paragraph (dealing with all these points) has been added to the revised version
There are some small language issues which, when corrected, will make the manuscript more easily readable. Throughout the text there are some places in which the wrong verb conjugations have been used (do/does, etc.) and some nouns lack their respective articles before them
Response
Thank you for your comment, a double check of English errors has been performed and corrections were done where appropriate.
Within tables 2 and 3, the frequently used phrase 'significantly improve/decrease/increase' should be corrected to the right tense ("improves") or changed from verb to noun (significant improvement, etc.) as used at other places within the same tables.
Response
Thank you for your comment, correction done as suggested
Reviewer 2 Report
Ammar et al. perform a systematic review and meta-analysis of polyphenol-rich interventions on both cognitive functioning and various brain health-based biomarkers. The study is well performed, well written, and the Authors did consider proper methodological aspects before conducting the analysis. The work is of interest to J Clin Med readers and to wider cognitive gerontology and aging fields. That being said, my major concerns include a potential bias in the interpretation of the results and a greater focus on positive sub-study cohorts during the discussion and conclusions.
Main criticisms:
- The study does significantly misinterpret the findings seen in results. Although none of the meta-analysis findings find a significant effect on any outcome, the Authors state in the conclusions and abstract as “This systematic review demonstrates that (poly)phenol-rich supplementation may improve some cognitive and cerebrovascular functions”. This does not correspond to either (CI -0.035 to 0.746, p=0.074 and CI -0.058 to 1.692, p=0.092) for TMT-A and B, respectively and even less so for BDNF or IL-6.
- Furthermore, despite the meta-analysis, the Authors isolate studies with a significant finding and high dosage that would “corroborate” their conclusions. I agree that one high dose sub-population did perform better compared to the remaining cohorts, however such findings in 30 high-dose subjects warrant greater reservation.
- What did the Authors consider as an aging population when performing the E/I criteria? That being said, was there an operational definition of “aging” (age cut-off?) that has been considered through the study?
- The exclusion criteria and the comparator classifications should be slightly adjusted. Some studies were excluded due to lack of PLA but others were included despite the cross-over design or had different dosages (example Mastroiacov et al.), thus violating the aforementioned exclusion criteria. Similarly, cross-over studies do not have PLA groups. This was further discussed in the results section, describing studies without PLA groups.
Minor/editorial comments:
Page 4, line 146 and onwards: The first two sentences are repeating from section 2.1.
Page 2, line 58: “One set of ARCD’s factors seems related to oxidative stress leading to neuroinflammatory process in neuronal cells” – please provide a reference
Double-check tables 2 and 3 for missing information. If data regarding the course of administration (acute, chronic) or the number of cases per PLA vs. treatment is missing, this should be specified.
Author Response
Reviewer #2:
Ammar et al. perform a systematic review and meta-analysis of polyphenol-rich interventions on both cognitive functioning and various brain health-based biomarkers. The study is well performed, well written, and the Authors did consider proper methodological aspects before conducting the analysis. The work is of interest to J Clin Med readers and to wider cognitive gerontology and aging fields. That being said, my major concerns include a potential bias in the interpretation of the results and a greater focus on positive sub-study cohorts during the discussion and conclusions.
The authors would like to thank the reviewer for the insightful and constructive comments on our work. We have carefully considered all of the suggestions and have revised the manuscript accordingly. We believe that our manuscript is much stronger as a result of making these modifications.
Please find below the authors’ responses to the individual comments.
Main criticisms:
The study does significantly misinterpret the findings seen in results. Although none of the meta-analysis findings find a significant effect on any outcome, the Authors state in the conclusions and abstract as “This systematic review demonstrates that (poly)phenol-rich supplementation may improve some cognitive and cerebrovascular functions”. This does not correspond to either (CI -0.035 to 0.746, p=0.074 and CI -0.058 to 1.692, p=0.092) for TMT-A and B, respectively and even less so for BDNF or IL-6.
Response
Thank you for raising this point.
The reviewer is right that the present meta-analysis results failed to provide evidence regarding the beneficial effect of (poly)phenols supplementation on cognitive performance during the trail making test (TMTA and TMTB) and also failed to provide evidence regarding the neuroprotective and anti-inflammatory effect of (poly)phenols supplementation. This interpretation is clear in both abstract and conclusion section and we highlighted it in the revision.
However, it should be noted that the systematic review with the recapitulative table 1 and 2 have focused on a larger number of cognitive and cerebrovascular functions (not only those included in the MA) and reported that some individual studies showed significant beneficial effect of (poly)phenols supplementation on some of these brain functions (please see table 1 and table 2). Unfortunately, these beneficial effects were not confirmed using MA given the lack of data (lack of studies investigated the same test/parameters; this point has been already discussed in the added weakness paragraph)
Therefore, as the abstract and conclusion section should provide an overview of all individuals and MA results, it was suggested that (poly)phenol-rich supplementation may improve some cognitive and cerebrovascular functions. However, to avoid ambiguity we specified in the revised version if the interpretation correspond to the individual studies’ results or to the MA results.
Please see modification in the revised text.
Furthermore, despite the meta-analysis, the Authors isolate studies with a significant finding and high dosage that would “corroborate” their conclusions. I agree that one high dose sub-population did perform better compared to the remaining cohorts, however such findings in 30 high-dose subjects warrant greater reservation.
Response
Thank you for raising this point.
We agree with the reviewer regarding this point. Therefore, we highlighted in a new “Strength and Weakness” paragraph that given some methodological issues such as the small sample size of individual studies, the results of the present systematic review and MA must be interpreted with caution
What did the Authors consider as an aging population when performing the E/I criteria? That being said, was there an operational definition of “aging” (age cut-off?) that has been considered through the study?
Response
Thank you for your comment, a cut-off age has been added in the inclusion and exclusion criteria and in PICOS table.
Adults aged 55 years old and over were considered as aging population (https://www.ncbi.nlm.nih.gov/pubmed/11815703)
The exclusion criteria and the comparator classifications should be slightly adjusted. Some studies were excluded due to lack of PLA but others were included despite the cross-over design or had different dosages (example Mastroiacov et al.), thus violating the aforementioned exclusion criteria. Similarly, cross-over studies do not have PLA groups. This was further discussed in the results section, describing studies without PLA groups.
Response
Thank you for raising this point.
We agree with the reviewer that some slight adjustment is needed in the inclusion/exclusion section. The exclusion criteria have been adjusted as suggested.
Minor/editorial comments:
Page 4, line 146 and onwards: The first two sentences are repeating from section 2.1.
Response
Thank you for your comment, the repeated sentences have been deleted
Page 2, line 58: “One set of ARCD’s factors seems related to oxidative stress leading to neuroinflammatory process in neuronal cells” – please provide a reference
Response
Thank you for your comment, reference has been added as requested
Double-check tables 2 and 3 for missing information. If data regarding the course of administration (acute, chronic) or the number of cases per PLA vs. treatment is missing, this should be specified.
Response
A thorough check of table 2 and 3 was performed and “not mentioned” was added were data is missing (not mentioned by the author in the full text)